# Properties of Aliphatic Ligand-Based Metal–Organic Frameworks

**DOI:** 10.3390/polym15132891

**Published:** 2023-06-29

**Authors:** Pavel A. Demakov

**Affiliations:** Nikolaev Institute of Inorganic Chemistry, Siberian Branch of the Russian Academy of Sciences, 3 Akad. Lavrentieva Ave., Novosibirsk 630090, Russia; demakov@niic.nsc.ru

**Keywords:** metal–organic frameworks, coordination polymers, aliphatic ligands, breathing, adsorption, optical properties

## Abstract

Ligands with a purely aliphatic backbone are receiving rising attention in the chemistry of coordination polymers and metal–organic frameworks. Such unique features inherent to the aliphatic bridges as increased conformational freedom, non-polarizable core, and low light absorption provide rare and valuable properties for their derived MOFs. Applications of such compounds in stimuli–responsive materials, gas, and vapor adsorbents with high and unusual selectivity, light-emitting, and optical materials have extensively emerged in recent years. These properties, as well as other specific features of aliphatic-based metal–organic frameworks are summarized and analyzed in this short critical review. Advanced characterization techniques, which have been applied in the reported works to obtain important data on the crystal and molecular structures, dynamics, and functionalities, are also reviewed within a general discussion. In total, 132 references are included.

## 1. Introduction

Metal–organic frameworks (MOFs) represent a class of compounds extensively studied in modern inorganic and coordination chemistry, materials science, and related fields. The possibility of structure–property design, which is provided by a wide range of available metal centers and organic ligands, unveils an unlimited library of functional materials with tunable and controllable properties, e.g., catalytic [1], sorption [2,3], optical [4], sensing [5,6], magnetic [7], and others. Many papers are devoted to MOF studies each year and that shows the continuously growing interest from the scientific community. A development of fast and energy-efficient synthetic protocols, and the usage of readily available and cheap starting chemicals makes this class of compounds valuable for industrial uses [8,9,10]. Record values of specific surface areas, and high thermal and chemical stabilities are achieved for MOFs at the current stage. 

In general, the vast majority of MOFs are based on aromatic carboxylates [11,12] and N-donor ligands [13], which possess their own structural rigidity and, therefore, allow a more straightforward design and predictable functionalization of the polymeric lattice. Aromatic MOFs apparently outperform other MOF classes in the tailorability of topologies and porosities, due to a huge diversity of the available aromatic ligands. Useful electron properties, such as highly effective luminescence and conductivity may be achieved for aromatic ligand-based MOFs. In addition, a benzene ring may be involved in substantial non-covalent bonding by C–H…π contacts (both as a donor and an acceptor) and π…π contacts, providing strong host–guest interactions. Linkers containing aliphatic or alicyclic cores take a very special place in the chemistry [14,15] of coordination polymers as the presence of such linkers often leads to drastically different properties of the resulting compounds, e.g., optical properties, adsorption uptakes and selectivity, and thermal stability. Furthermore, flexibility of the aliphatic backbone may provide a unique type of conformation-controlled breathing [16], which is uncommon for purely aromatic-based MOFs. Such an obviously highly designable type of framework mobility leads to its increased variability and makes a sponge-like behavior [17] of the sorbent possible for selective separations and storage applications. 

Despite some specific advantages of aliphatic/alicyclic carboxylic acids, e.g., their availability from the renewable natural sources instead of hydrocarbon feedstocks, coordination networks based on such ligands are far less common. Table 1 and Table 2 present the main aliphatic and alicyclic carboxylate bridges being used in the synthesis of coordination polymers, respectively. Their numbers of crystal structures, reported in the literature, are one/two orders smaller when compared to the terephthalic acid (see Table 1)—the simplest aromatic dicarboxylate ligand. These numbers clearly illustrate a low spread of saturated bridging carboxylates compared to their aromatic parents in MOFs. The chemistry of aliphatic-based coordination polymers is not well-developed yet, and only a few reviews are devoted to a brief systematization of their properties [15,18,19]. In particular, the conformational lability of such ligands may complicate the self-assembly of a regular periodic coordination lattice and may tangle the further characterization of the samples. Especially, an application of standard X-ray diffraction techniques often appears more difficult, owing to the possible conformational breathing and subsequently increasing disorder in sample crystal structure. Nevertheless, specific properties such as high hydrophobicity, low photoactivity, and controllable conformation dynamics being driven by the aliphatic backbone nature make such types of metal–organic frameworks promising in photochemistry and photocatalysis, in the design of uncommon adsorption properties, and the synthesis of stimuli-responsive smart materials [20], the latter possessing structures and/or properties that are switchable under external impact.

Owing to the specific chemical nature of aliphatic-backboned linkers, their rareness in MOF chemistry, and some difficult aspects in their synthesis and behavior, this critical review is devoted to a comprehensive consideration of unique properties inherent to metal–organic frameworks based on purely (or almost purely) aliphatic, including alicyclic, ligands. To present and discuss the properties led by the presence of the saturated hydrocarbon backbone more clearly, several ligand types which comprise such fragments as only a part of their molecular structure or contain heteroatomic functional groups, were consciously excluded from the present consideration. Therefore, this work is mainly focused on MOFs containing the most common saturated dicarboxylic acid anions and several quite unique purely aliphatic struts, e.g., 1,4-diazabicyclo [2.2.2]octane (dabco) derivatives. Advanced characterization techniques, which have been applied in the investigations of these types of MOFs, are discussed in relation to the corresponding properties. In the conclusion part, the main achievements in the field of aliphatic-based MOFs are generalized, and further prospects and unresolved problems are highlighted.

## 2. Properties of Aliphatic-Based MOFs

### 2.1. Conformational Breathing and Methods for Its Characterization

Traditionally, single crystal X-ray diffraction analysis (SCXRD) represents the most common technique for the structural determination of coordination compounds, including MOFs [22,23]. This is a quite simple and straightforward method which makes possible an unambiguous and highly accurate determination of atomic coordinates, intermolecular contacts, and packing effects in the solid state. However, the dynamics of a polymeric coordination network often limit the capability of SCXRD due to the poor long-range order with a large amount of defects introduced and easy fragmentation of large single crystals. In the case of aliphatic-backboned ligands, often bearing conformational dynamics, such difficulty in the preparation of high-quality single crystals of MOFs is even more noticeable, especially when quite high energy barriers separate the local energy minima representing metastable conformers of the ligand. 

However, in some cases the single-crystallinity of a conformationally flexible sample may be retained upon activation or guest exchange, which makes it possible to use SCXRD for the structural determination of multiple phases of MOF. Most reported examples are related to the glutarate chain deformation, which represents an example of quite soft deformations appearing between several stable forms of the bridge with low energy barriers. 

#### 2.1.1. Breathing of Glutarates

A flexibility of the glutaric (pentanedioic; H_2_glu) acid anion is mainly attributed to the chain rotation around several C(sp^3^)–C(sp^3^) bonds. Although such types of rotation represent quite free motions when any torsion angle value from 0 to 180° is not forbidden, three main metastable conformers may be deduced for glutarate from the sterical point of view, i.e., *gauche*–*gauche*, *anti*–*gauche* and *anti*–*anti* rotamers (Figure 1).

Several works report single-crystal XRD structures of activated (desolvated) phases of glutarate-based MOFs. In these structures, the aliphatic chain of glutarate adopts *anti*–*gauche* conformation representing a bended linker [24,25,26]. Despite the existence of shorter angular *gauche*–*gauche* conformation, its occurrence in the activated structures seems to be elusive for the considered frameworks due to the sterical hindrance between metal blocks interconnected by such a strongly entangled linker. Moreover, no framework opening driven by an *anti*–*gauche* → *anti*–*anti* transition was reported for these MOFs upon any guest inclusion. 

An example of a pronounced framework contraction driven by the rotation of the glutarate aliphatic chain was reported in [27] and further in [28]. In the latter work, for MOFs had the initial formulas [Cu_2_(glu)_2_(bpa)]·2.4H_2_O (**1a**; bpa = 1,2-bis(4-pyridyl)ethane) and [Cu_2_(glu)_2_(bpp)]·2acetone (**2a**; bpp = 1,3-bis(4-pyridyl)propane). Upon partial desolvation, a +*gauche* → -*gauche* transition was observed in [Cu_2_(glu)_2_(bpa)]·0.93H_2_O (**1′a**), according to SCXRD. Since the similar desolvation of **2** led to a polycrystalline material, an acetone exchange to CO_2_ in supercritical conditions was applied to retain the single crystallinity of the framework, and a similar conformational transition was then successfully proven for [Cu_2_(glu)_2_(bpp)]·8.2CO_2_ (**2′b**). Subsequent complete activation did not lead to any significant changes in the framework geometries of [Cu_2_(glu)_2_(bpa)] (**1′**) and [Cu_2_(glu)_2_(bpp)] (**2′**), and it was then proven by PXRD-refined structures. Despite the retaining of main *anti*–*gauche* structural motif, such enantiomeric interconversion led to a more pronounced bending of an angular glutarate linker (Figure 2), which resulted in a shrinkage of solvent accessible volumes by 44% **2**.

As it was discussed above, breathing of the polymeric crystal structure often affects strongly the quality of single crystals and decreases the overall crystallinity of the sample. In this regard, high-resolution powder X-ray diffraction (PXRD) becomes very important as a method of structural determination of breathing MOFs [29,30,31]. Since the quality of data, available on laboratory powder diffractometers, often appears to be not enough for the direct structure solution, in the most difficult cases this method is appended by theoretical modeling, DFT computations, electron diffraction [32,33,34,35], or other advanced techniques which allow finding an initial structural model to be further refined for polycrystalline samples on the basis of PXRD data. However, even if PXRD structure refinement is impossible or unnecessary, the qualitative analysis of powder data represents an easy and express method for the observation of breathing behavior and elucidation of the driving factors, such as solvent, temperature, and pressure. Several works report the PXRD observation of breathing in glutarate-based MOFs which, however, also contain another flexible ligand or bear a mobile layered structure [36,37]. The absence of both the quantitative PXRD data interpretation and other experimental data did not allow to attribute the structural transition solely to the glutarate flexibility in these cases. 

An unusual example of breathing in glutarate-based MOF while the main change in the ligand conformation does not significantly affect the aliphatic backbone, was reported by Serre and coworkers [38]. A compound with the formula [Ca(glu)] called bioMIL-2 was synthesized solvothermally in DMF. Upon hydration, a fast transition to a new phase [Ca(H_2_O)(glu)] (bioMIL-2-hyd) was observed. The hydration was accompanied by the coordination of water molecules to Ca^2+^ coupled with the change the coordination type of one carboxylic group and the subsequent change in the topology of the coordination framework. De-coordination of the previously bridging COO-group led to its pronounced rotation around the C(sp^3^)–C(COO) bond (Figure 3), which is quite rare for the unsubstituted aromatic carboxylates and also provided by the conformational flexibility of the aliphatic dicarboxylate. 

#### 2.1.2. Breathing of Adipates

Adipate as an anion of adipic (hexanedioic, H_2_adp) acid represents a longer glutarate analogue, which has a higher degree of structural mobility and subsequently a larger number of possible rotamers. However, twice-twisted conformations (*anti*–*gauche*–*gauche* or *gauche*–*anti*–*gauche*) were evidenced to be the most stable in a series of adipate-based MOFs in the work by Moon et. al. [39]. 

Solid-state nuclear magnetic resonance (NMR) spectroscopy is a powerful tool for analyzing the structural dynamics of metal–organic frameworks. Although a large number of solid works, which concern investigation of MOFs by means of ^2^H NMR spectroscopy, have been published [40,41,42,43,44], such methods, to the best of our knowledge, have not been applied to the aliphatic dicarboxylates in MOFs, possibly due to the low availability of the deuterated forms of such ligands. Similarly, ^129^Xe NMR spectroscopy [45,46] has not been applied to MOFs containing such ligands yet. However, by using ^13^C NMR, it is possbile to distinguish diverse conformations of the aliphatic ligand backbone [47,48] and the obtained information appears very important to support poor structural data for the samples with low crystallinity.

Solid-state NMR on the metal nuclei also becomes a powerful tool to analyze the dynamic behavior of metal–organic frameworks [47,49,50]. Though it mainly images the changes in the metal coordination environment, the combination of metal nuclei-NMR with the structural modeling appears to be an effective instrument for the deep understanding of phase transitions and for further confirmation of the structural changes on an atomic level in poorly crystalline samples. As an obvious and significant drawback of all the NMR methods, one can single out their applicability only to compounds based on diamagnetic metal centers. A massive combination of NMR and theoretical methods were applied by Stock and coworkers in the characterization of a breathing aluminum-adipate MOF based on the linker with a high degree of conformational diversity [51].

In that work, the authors synthesized an adipate-containing [Al(OH)(adp)] belonging to the well-known **MIL-53** family of metal–organic frameworks, which are based on metal carboxylate-hydroxyl chains. The structural dynamics of [Al(OH)(adp)] appeared to be strikingly different from its terephthalate-contained parent [Al(OH)(bdc)] (H_2_bdc = terephtalic acid), whose dynamics were earlier described as a combination of the coordination environment deformation and rigid linker bending [14,18]. The as-synthesized hydrated form [Al(OH)(adp)]·H_2_O contains channels about 3.2 Å in diameter lying across the walls constructed from two types of adipate ligands adopting *gauche*–*anti*–*gauche* (half of ligands) and *gauche*–*anti*–*anti* (another half) conformations. These channels are occupied by water (Figure 4). Upon activation of the compound, the shift of atom coordinates in the Al coordination environment is insignificant (3.5 times lower than similar values for [Al(OH)(bdc)]), however, the aliphatic backbone of the half of adipate ligand bends from a zigzag *gauche*–*anti*–*gauche* to an arcuate *gauche*–*anti*–*anti* shape with a subsequent formation of a densely packed structure. Interestingly, this rearrangement leads to the notable elongation of the aliphatic strut for ca. 0.4 Å, though it results in a loss of the framework porosity, what is apparently attributed to high hydrophobicity of the long alkane chain in the adipate. 

Poor crystallinity of highly flexible [Al(OH)(adp)] samples allowed determination of the crystal structure for only a dense activated form. The structure of the hydrated form was suggested using the combination of molecular modeling, force-field optimization, and DFT computations with further support by qualitative PXRD analysis, ^13^C–, and ^27^Al solid-state NMR data. The dense form does not adsorb nitrogen at 77 K; however, its contraction is fully reversible as its expansion appears at 60% relative humidity at room temperature. Desorption of water vapor, accompanied by the transition of the framework to a nonporous form, is observed at a relative humidity of about 25% (Figure 5).

#### 2.1.3. Breathing of *Trans*-1,4-Cyclohexanedicarboxylates

Cyclohexane-1,4-dicarboxylic acid (H_2_chdc) is an industrially produced compound which is mildly spread in MOF chemistry. The cyclohexane ring is known to present mainly in a chair conformation and three diastereomers exist for H_2_chdc, namely (*e*,*a*)-H_2_chdc, (*e*,*e*)-H_2_chdc, and (*a*,*a*)-H_2_chdc (Figure 6), the last two of which are rotational conformers. While the (*e*,*a*) isomer (*cis*-cyclohexane-1,4-dicarboxylate) is an almost rigid angular linker, (*a*,*a*) and (*e*,*e*) conformers of *trans*-cyclohexane-1,4-dicarboxylate, have only a ca. 12 kJ/mol energy barrier between them, estimated in solutions [52]. These provide a mobile system and are able to interconvert between each other even in a polymeric coordination network. Such type of flexibility represents an example of semi-rigid behavior with no free drift of the framework parameters. Since such rotational interconversion of *trans*-chdc is quite inhibited, there are several reports for breathing in chdc-based MOFs by its bending mechanism similar to the bending of the aromatic ligands with rigid cores [53,54]. However, several notable examples of conformation-controlled breathing in *trans*-1,4-chdc MOFs with the structural characterization of the framework isomers will be discussed below. 

An example of (*e*,*e*) ↔ (*e*,*a*) interconversion during the solvent inclusion/activation was reported by De Vos et. al. [55] for [Zr_6_O_4_(OH)_4_(chdc)_6_] MOF, a chdc-constructed analogue of the terephthalate-based UiO-66. The as-synthesized UiO-66-chdc contains only (*e*,*e*) ligand conformation, what makes its unit cell parameters very similar to the parent [Zr_6_O_4_(OH)_4_(bdc)_6_]. A solvent-accessible volume in this structure is 43%, showing high porosity of this framework. However, [Zr_6_O_4_(OH)_4_(chdc)_6_] collapses into a non-porous X-ray amorphous phase upon evacuation, which could be easily returned into the as-synthesized form while immersed in several solvents, i.e., water, ethanol, or DMF, representing an example of reversible breathing. Although the structure of the collapsed phase was not directly determined, solid-state MAS ^13^C-NMR revealed 1/3 of the bridging ligands to adopt the biequatorial (*e*,*e*) conformation, and the remaining 2/3 to adopt the shorter, biaxial (*a*,*a*) form. According to these data, the structure of the collapsed phase was modelled theoretically to undergo a tetragonal distortion (Figure 7) of cubic UiO-66-like cages. The framework amorphization during such a transition was explained to occur due to the high symmetry of these cages and the resulting presence of several equivalent directions of collapse, randomly distributed in a three-dimensional coordination lattice within the range of no more than ca. 10 unit cells.

An example of inverted (*e*,*a*)–(*e*,*e*) interconversion, while the narrower pores present in the as-synthesized form, was reported for [Al(OH)(chdc)] denoted by the authors as Al-CAU-13 [56]. This framework represents an analogue of the well-known MOF [Al(OH)(bdc)] (MIL-53(Al)) with terephthalate anions being substituted by flexible *trans*-1,4-cyclohexanedicarboxylates. Similar to the parent one and to the aluminum adipate described above, this structure contains one-dimensional channels lying between polymeric Al-hydroxide chains interconnected by dicarboxylate linkers. In the structure of CAU-13, these channels have a distorted shape due to the presence of two conformations of the chdc^2−^ bridge (Figure 8a), which differ in lengths and are situated along different crystallographic directions.

The activated form of [Al(OH)(chdc)] was found to be similar to the as-synthesized framework, except some alignment of metal coordination environment with a change of Al–O bond length intervals from 1.79(3)…2.07(2) Å in the “aqueous” form to 1.82(3)…1.93(2) Å in the activated form. The unit cell volume of CAU-13 increased by only 2% upon activation, which indicates the absence of significant rearrangements (e.g., channel opening), typical to its terephthalate prototype MIL-53. In contrast, expansion of CAU-13 with (*e*,*a*)–(*e*,*e*) transition occurred at the adsorption of isomeric xylenes from both vapor and liquid phases. Such a transition results in the formation of more symmetric rhombic channels (Figure 8b) and some distortion of the Al^3+^ environment. For example, Al–O bond lengths in the adduct with ortho-xylene range from 1.85(4) Å to 1.99(4) Å. A provided data on interatomic distances in metal-carboxylate chains allows to suggest that the distortion of the Al^3+^ environment increases depending on the polarity of the adsorbate in the series: activated (no guest) < xylenes < water and is apparently driven by the weak intermolecular contacts between guest molecules and OH groups of the metal-carboxylate chains. A complete reversibility of the conformational transition in [Al(OH)(chdc)] was proven by the authors. A further reported study on the adsorption of polymethylsubstituted pyrazines [57] also revealed the size- and shape-dependency of Al-CAU-13 breathing. 

Electron paramagnetic resonance (EPR) is also a very sensitive method for which the high crystallinity of the sample is not required. This method involves the study of paramagnetic centers in various components of the metal–organic framework: metal centers [58,59,60], guests [61,62,63], and organic bridging ligands [64,65,66], where, in the absence of their own paramagnetism, the corresponding functionalization may be carried out. Doping of a diamagnetic MOF lattice by trace amounts of paramagnetic centers in a metal node (for example, by partial substitution of diamagnetic Zn^2+^ cations for Cu^2+^) or within pores (for example, by introducing a stable nitroxide radical as a paramagnetic probe) should be singled out, as a high dilution of the paramagnetic signal source makes it possible to study the behavior of an almost unperturbed MOF structure where local changes around the dopant do not affect the overall structure and bulk properties of the polymeric network, including its breathing. In particular, structural rearrangements in ZIF-8 [67], frameworks of the MIL-53 family [68,69], and the [Zn_2_(bdc)_2_(dabco)] (DMOF-1) family [59,70,71] have been extensively studied by diverse EPR techniques applied to the diluted paramagnetic centers. 

An example of (*e*,*a*)–(*a*,*a*), interconversion of *trans*-1,4-cyclohexanedicarboxylate ligand in the coordination framework during the solvent exchange and activation, when EPR measurements data were used to investigate the breathing of poorly crystalline sample, was presented for [Zn_2_(chdc)_2_(dabco)] (dabco = 1,4-diazabicyclo [2.2.2]octane) in [72]. This MOF is isoreticular to its well-known terephthalate-based parent [Zn_2_(bdc)_2_(dabco)] (DMOF-1). It is constructed from {Zn_2_(OOCR)_4_(N)_2_} paddle-wheel units, which are six-connected and bound to each other by rigid linker dabco and semirigid linker chdc^2−^ to form a distorted three-dimensional primitive cubic (pcu) network (Figure 9). The intersecting channels, lying across three different directions, provide a solvent accessible volume of 43% in the as-synthesized form. Its activation leads to the structure collapsing into a new phase with poor crystallinity. However, the structure was found to be fully restored to the initial crystal structure at NMP solvent treatment. Attempts to determine the crystal structure of the collapsed form were unsuccessful, however, ENDOR investigation of close-to-metal hydrogen environment performed with Cu-doped samples of [Zn_2_(chdc)_2_(dabco)] has proven the existence of only (*a*,*a*)-chdc conformation in it. Further investigation of benzene and cyclohexane adsorption from the vapor demonstrated multi-stepped hysteretic sorption and moderate benzene/cyclohexane selectivity applicable to the separation of these substrates. 

#### 2.1.4. Breathing of Other Aliphatic MOFs

Structural features and adsorption properties of two Zr(IV) metal–organic frameworks, namely Zr-fumarate and Zr-succinate were investigated in [73]. Succinate (butanedioate) in its longest *anti* conformation and fumarate (*trans*-buten-2-dioate; Figure 10a) represent a pair of bridges with similar geometry and quite similar molecular lengths. The main difference occurring between these bridges is a conformational lability of a succinate aliphatic chain which could introduce perceptible conformational dynamics to the coordination framework, instead of the rigid backbone of fumarate. Zr-succinate was found to possess significantly lower CO_2_ and N_2_ adsorption uptakes, which was attributed to the swelling of the flexible MOF samples upon activation. However, its CO_2_/N_2_ selectivity, which was calculated in terms of ideal adsorbate solution theory (IAST), demonstrate a pronounced increase with the corresponding selectivity values ranging from ca. 1.5 times at 0 mbar to more than 2-fold at 100 mbar. This phenomenon was attributed to the reversibility of the swelling in Zr-succinate (Figure 10b) and, therefore, its better fitting to the highly polarizable CO_2_ molecules, representing some sponge-like behavior of the coordination framework. 

Ortiz et al. reported the pressure-induced dynamics for two MOFs based on butane-1,4-diphosphonate and hexane-1,6-diphosphonate ligands [74], which can be recognized as phosphonate analogues for adipate and suberate, respectively. At high pressure, both compounds demonstrated a proton transfer from the coordinated phosphonic acid group to guest water molecule and subsequent charge separation. In addition, the suberate-bridged framework at high pressure underwent a pronounced contraction driven by coiling of the long aliphatic chain in the bridge from almost pure *anti*-conformation to the chain containing two *gauche-* entanglements (Figure 11). The adipate-bridged framework did not reveal a similar transition, apparently due to low (insufficient) chain length which does not allow the contraction due to the sterical hindrance between the metal nodes in the framework. The structural transitions were characterized by single-crystal X-ray diffraction, and theoretical calculations on stability enthalpies were performed for different phases to support the experimental observations. The unique piezo-mechanical response shown for Zn-hexane-1,6-diphosphonate revealed great perspectives of the soft aliphatic MOFs in the preparation of materials with negative linear compressibility (NLC) and other unusual mechanical properties. 

As it was shown above, an ability to soft deformations in the coordination framework could naturally be an important factor for the retention of high quality of single crystals during the structural transition. Such a feature could be very useful while carrying the photoinduced reactions between the photoactive substrates isolated into the coordination framework. Typical photoreactions, e.g., photoinduced [2+2]-dimerization, obviously lead to some change (typically to decrease) in the summary molecular volume of the reacting building blocks and, therefore, may lead to considerable structural rearrangements. In [75], the conformational flexibility of the coordinated allylmalonate anion promoted the UV-induced [2+2] cycloaddition between C=C double bonds of the allylmalonate and bis-4-pyridylethylene acted as a second bridge in the coordination framework. In the crystal structure of the as-synthesized compound before irradiation, the mentioned bonds are located quite far from each other. However, some low-energy rotations of allyl substituent around the C(sp^3^)–C(sp^3^) bond obviously occur and make the closer approaching of double bonds possible, suitable for their intermolecular coupling. The SCXRD for the crystal after the irradiation revealed a partial dimerization (Figure 12) which was thus driven by the flexibility of the aliphatic allylmalonate ligand. Despite this, it would be fair to note that in [76] and some other works [77,78,79,80], the ligand flexibility is evident to be not strictly required for the successful carrying out of UV-induced reactions in MOFs.

### 2.2. Adsorption Properties

Since the discovery of the first examples of permanently porous metal–organic frameworks [81,82,83], porosimetry under equilibrium conditions between the condensed and vapor phases for an adsorptive has become the main method of their textural characterization. The most common adsorptives for such measurements are nitrogen (T_boil_ = 77 K), argon (T_boil_ = 87 K), and carbon dioxide (T_subl_ = 195 K). Such a method makes it possible to effectively evaluate fundamental characteristics of porous structures, such as the specific surface area of the sorbent, pore volume, type and geometry of cavities, their size distribution, etc. Usually, the results of sorption measurements on well-prepared and structurally rigid samples fit well to the data calculated from XRD-derived crystal structures for these sorbents. 

In addition to textural characterization, the investigation of volumetric adsorption of different gases and volatile organic substance vapors are of great practical importance, due to several preferences of guest-responsive frameworks for the adsorption [18,84]. Their guest-dependent mobility, which often leads to a more effective fitting of the host pore size and geometry to the guest molecules (sponge-like behavior), opens the possibility of enhanced adsorption selectivity and stability of the guest, adsorbed into porous sorbent, for its prolonged storage. In practice, such stabilization is often expressed in the phenomenon of hysteresis between the desorption and adsorption processes [36,85,86], when the adduct obtained after the pore opening in MOF at a certain pressure appears to be stable under a significantly decreased pressure without loss of the adsorbate. Again, the structural dynamics of the ligand aliphatic backbone, which is one of the possible ways to effectively increase the adjustability of the host sorbent to the guest while adsorbed, represents a so-called sponge-like behavior of the corresponding frameworks. 

The saturated backbone of aliphatic dicarboxylates or similar ligands does not contain atoms with any pronounced electron-donor or electron-acceptor properties. The absence of strong adsorption centers, as well as the low polarizability of such a core, implies that the adsorption properties of aliphatic MOFs for typical polar or easily polarizable adsorbates can be mainly related to other framework components, for example, open metal cites. However, the inert saturated backbone possesses strong hydrophobicity which could reveal the adsorption properties and selectivity to be drastically different from the conventional adsorption sites [87,88,89]. A strong impact of hydrophobicity on the adsorption properties has been noted even for methyl substituents in typical π-conjugated ligands, which was expressed in a poor or stepped uptake of highly polar substrates, such as water [90,91,92,93] (Figure 13). 

Despite the C(sp^3^)–H bond being poorly polar to be a conventional adsorption center, the host–guest intermolecular interactions with these bonds may be enhanced significantly due to their multiplicity. Moreover, a high hydrophobicity of these aliphatic or alicyclic moieties may provide a certain affinity to the non-polar substrates, such as hydrocarbons. Since the main alicyclic dicarboxylate linkers, such as chdc^2–^, bicyclo[2.2.2]octane-1,4-dicarboxylate, and cubane-1,4-dicarboxylate [15] are very close to the aromatic terephthalate (benzene-1,4-dicarboxylate, bdc^2–^) in terms of their linear geometry and similar molecular lengths, their implementation in porous metal–organic frameworks as single bridges or in the mixtures with bdc^2–^ approaches a very fine tuning of pore size as well as modulating the nature of host–guest interactions, which are strongly different between aliphatic and aromatic ligand struts. 

A pronounced impact of the aliphatic backbone on the adsorption selectivity was reported for [Zn_4_O(cdc)_3_] (CUB-5; cdc^2–^ = cubane-1,4-dicarboxylate) [94], which is an isoreticular analogue of the well-known MOF-5 [Zn_4_O(bdc)_3_], where bdc is terephthalate (Figure 14). Similar to its aromatic parent, CUB-5 has a highly symmetric cubic pcu coordination lattice and contains voids about 11 Å in diameter connected by windows with an aperture of about 7 Å. The cdc^2–^ aliphatic backbones act as walls of the channels, and the C(sp^3^)–H bonds are oriented directly into the channels.

CUB-5, when compared to MOF-5, possesses a slightly lower overall porosity and comparable saturation uptake for various hydrocarbons (benzene, cyclohexane, and several C_6_ alkanes), but adsorption patterns for these MOFs differ strongly. For both compounds, stepwise adsorption isotherms were observed, but, for example, a pressure of about 0.25 kPa for MOF-5 and less than 0.1 kPa for CUB-5 was needed for the abrupt increase (step on the isotherms) in the adsorption uptake (Figure 15). Therefore, at low pressures, the hydrocarbon adsorption on CUB-5 is significantly higher than for MOF-5, especially for the benzene adsorption. DFT calculations results showed that the C(sp^3^)–H bond becomes the main adsorption site for benzene in CUB-5, and the heat of C_6_H_6_ adsorption CUB-5 is 7 kJ/mol higher than that value for MOF-5. The authors proposed an involvement of aliphatic ligands to be a potential method for improving the adsorption characteristics of highly porous MOFs and further implemented a similar strategy in a more complex work [95]. 

In a work by Yang and coworkers [96], a three-dimensional porous MOF [Cu_2_(bcpdc)_2_(dabco)], called as ZUL-C3 by the authors, was obtained. This structure is based on bicyclo [1.1.1] pentane-1,3-dicarboxylic acid (H_2_bcpdc), a very rare ligand whose molecular structure, however, fully falls into the family of linear dicarboxylates with polycycloalkane cores. This coordination framework contains typical {Cu_2_(OOCR)_4_(N)_2_} paddle-wheel blocks, interconnected by dabco and bcpdc^2–^ linkers into a DMOF-1-type distorted primitive cubic structure. Hydrogen atoms of alicyclic fragment C–H bonds are arranged directly into the channels (Figure 16a), revealing the possibility of strong non-aromatic type intermolecular interactions. A thorough study of the adsorption of isomeric xylenes and ethylbenzene was carried out for ZUL-C3, including vapor and liquid competitive adsorption, breakthrough experiments, SCXRD of the guest-loaded samples, and DFT simulations. A preferable adsorption of *o*-xylene was observed, resulting in *o*-xylene: *p*-xylene selectivity for up to 22 factor and up to 8 factor for the other two isomers. These experimental results were supported by the adsorption enthalpy calculations, which revealed the strongest binding of the *ortho*- isomer to the ZUL-C3 framework. Moreover, DFT theoretical computations revealed plenty of intermolecular host–guest contacts, out of which the interactions between guest CH_3_ groups and bicyclo [1.1.1]pentane ligand cores are apparently the strongest (Figure 16b), providing the best fitting of the *o*-xylene molecules to the host pore geometry and arrangement of poorly polar adsorption centers. 

Another way of using (poly)alicyclic dicarboxylates in the design of selective adsorbents exploits a consideration of cubane and similar cores as quite bulky struts, able to modulate the effective size of micropores. No matter to the specific host–guest interactions, such moieties can be recognized as three-dimensional pavers of the voids instead of typically planar and subsequently two-dimensional benzene cores in the aromatic ligands. In the case of the latter, an obligatory rotation of the aromatic rings leads to an effective size of the channels and apertures to be highly dispersed within a regular crystal structure and moreover to be possibly drifted under the impact of relatively small changes in the working conditions, e.g., temperature, pressure, humidity, and others. Changing the aromatic channel pavers to the cubane-like moieties, possessing their own three-dimensional bulkiness, allows diminishing the influence of the ligand rotation for fixing the effective channel or aperture sizes. Therefore, a largely more strict and predictable sieving effect becomes possible in the related porous structures. In recent studies, such a “bulky polyalicyclic” approach was successfully applied in the Kr/Xe separations [97,98], light hydrocarbon purification [99,100], separation of xylene isomers [100], and selective uptake of linear and lower-branched hexane isomers [101].

### 2.3. Optical Properties

The aliphatic moieties are known to absorb UV and visible light very weakly due to the absence of any π-systems and any other highly light-absorbing molecular moieties. This feature of the ligand could be naturally transferred to the coordination framework constructed from such ligands if corresponding metal centers, which are also transparent, are used. Metal cations with fully filled or empty valence electron shells (e.g., d^0^ or d^10^) could be suitable for this purpose. Such a proposal has been confirmed for several Zn(II)- [102,103,104] and La(III)-based MOFs [105] containing *trans*-1,4-cyclohexanedicarboxylate, bicyclo[2.2.2]octane-1,4-dicarboxylate, or 1,4-diazabicyclo[2.2.2]octane-N,N′-dioxide. The reported diffuse reflection (DR) spectra for such compounds have unambiguously proven their very high transparency in UV/vis, at least up to 310 nm. In [102], a porous MOF [Zn(bcodc)]·0.75DMF (TMOF, Figure 17a) was further used as a dilution matrix for guest pyrene molecules, and gave the absorption spectra of the pyrene@TMOF adduct to contain well-resolved pyrene absorption bands, mainly typical of its highly diluted solutions, but not typical for pyrene in the solid-state (Figure 17b). The authors concluded that UV-transparent porous MOFs can be applied as perfect matrices for spectroscopic studies of isolated (individual) photoactive guest molecules.

Luminescence is one of the most frequently studied properties of MOFs. The luminescent properties of coordination compounds in the solid-state depend on their chemical composition and photoactivity (i.e., electronic structure, absorbance, appearance of emissive excited states, etc.) of their constituents. A mutual arrangement of photoactive centers in the crystal structure is also known to have a strong impact on the luminescence characteristics due to the dependence of photoexcitation energy transfer mechanisms and their efficiencies on the packing of the related atomic and molecular blocks. By understanding the electronic structure of organic molecules as well as analyzing great amount of reported experimental data, one can unambiguously state that the saturated aliphatic backbone participates insignificantly in the energy transfer processes in luminescence. The photoactivity of the aliphatic carboxylate ligand may be related to the presence of the RCOO carboxyl group, to the charge transfer processes by either MLCT or LMCT [106,107], and to dipole–dipole Förster resonance energy transfer (FRET) mechanisms. For those compounds where the coordinated carboxyl group of the aliphatic carboxylate was reported as the origin of luminescence, a blue or violet emission was typically observed [108,109,110,111].

In this regard, low photoactivity of the aliphatic backbone could be applied in a manner reversed to that described above: to dilute photoactive centers bound into the polymeric coordination framework for improvement in its stability to photodegradation or for fine-tuning the emission color and other characteristics of the emissive blocks interconnected by transparent aliphatic struts [112,113]. The hydrophobicity of the saturated ligand core should also be mentioned as a factor which possibly increases the stability of luminophore to hydrolysis, obviously a very important property in terms of practical relevance.

A family of Pb-halide succinates, adipates, and *trans*-1,4-cyclohexanedicarboxylates was described in a series of reports [114,115,116]. These compounds are based on cationic lead–halide coordination networks interconnected by aliphatic dicarboxylate bridges into 2D or 3D polymeric structures. In a representative example [116], the luminescence of two similar MOFs with the formulas [Pb_2_Cl_2_(chdc)] and [Pb_2_Br_2_(chdc)] was studied in detail. The emission spectra of the chloride-containing compound demonstrated a two-component character, and n-π* transitions in the carboxyl group were found to influence the luminescence. Varying the excitation wavelength led to fine tuning the luminescence color. In particular, the authors achieved a near-white emission with a quantum yield of 20% at hard UV excitation (λ_ex_ = 304 nm). In the case of a bromide compound, the emission of the COO group was silent, which was explained by the different geometry of lead(II) coordination. White emission with quantum yields up to 17.2% at soft UV excitation (λ_ex_ = 354 nm) were achieved for [Pb_2_Br_2_(chdc)]. Both presented emission efficiencies have outperformed the record values previously reported for anionic perovskite-like haloplumbates. Moreover, the photostability of these MOFs was found to be drastically higher in comparison the known perovskite-like anionic lead halide complexes, which was attributed to the impact of the long-chained and UV-transparent chdc^2–^ ligand, fully corresponding to the dilution principle described above. 

### 2.4. Ferroelectric and Paraelectric Properties

An almost free rotation around C(sp^3^)–C(sp^3^) single bonds results in a pronounced molecular rotation in aliphatic chains and, in some cases, in axial rotation of alicyclic fragments. Such a phenomenon was described earlier for the dabco ligand, bearing pure D_3h_ symmetry in the case of an unfrozen rotation freedom degree (Figure 18) and quazi-D_3h_ symmetry at the related vibrational disorder. Freezing of rotation motions at low temperature breaks D_3h_ symmetry to the lower ones, which allows structural transitions in a crystal, including induction of chirality [117]. Indeed, the frozen conformations of dabco are apparently chiral (D_3_(S) and D_3_(R); (see Figure 18), so such types of structural transition may induce a spontaneous polarization of the crystal, providing paraelectric or ferroelectric properties of the material as a consequence. 

Such observations were reported for two MOFs based on the 1,4-diazabicyclo[2.2.2]octane-*N*,N’-dioxide (odabco), namely [Ag_3_(odabco)(NO_3_)_3_]·H_2_O and [Ca(odabco)(H_2_O)_4_]Cl_2_ [118]. For the Ca-based compound, a tetragonal-to-orthorhombic order-to-disorder transition, both phases being paraelectric, was found at T ≈ 177 K. For the Ag-based compound a displacement-type phase transition from orthorhombic phase to monoclinic one, both ferroelectric, appeared at temperature decrease to T ≈ 217 K. The remnant polarization was measured to be ca. 0.23 μC/cm^2^. From the structural point of view, the phase-transition in [Ag_3_(odabco)(NO_3_)_3_]·H_2_O could be mainly attributed to the distortion of the odabco aliphatic core, which is expressed in a variation of the N-C-C-N dihedral angle (Figure 19), showing its achirality breaking. Similar spontaneous polarization effects were also reported for some odabco-based salts or co-crystals [119,120]. 

In a recent work, an antiferroelectricity appearing at a low-temperature phase transition was found for the compound [Zn_2_(bcodc-F_2_)_2_(dabco)] (H_2_bcodc-F_2_–2,2-difluorobicyclo[2.2.2]octane-1,4-dicarboxylic acid; dabco = 1,4-diazabicyclo[2.2.2]octane) [121] bearing two types of molecular rotors containing dabco-like cores. These linkers are situated in perpendicular directions of a pcu-type 3D polymeric lattice. At high temperatures, such moieties possess a rapid rotation, and no distinct conformations of the dabco core appear in the crystal structure. At T = 100 K, the rotation apparently freezes, which leads to a collective dipole ordering in the crystal, in which the dipole moments of the initially rotated linkers largely cancel each other. A DFT-performed energy scan for the enantiomeric twisting in a bcodc-F_2_ ligand showed only a ca. 3.5 kJ/mol energy barrier between the enantiomeric polar structures, which support a low temperature being necessary for such transitions to make the energy of the thermal motions lower than the found rotation barrier. 

### 2.5. Thermal Stability

Aliphatic moieties are known to possess a relatively low thermal stability compared to the cognate aromatic fragments. However, the aliphatics are characterized by lower carbon content and higher volatility, which often leads to a specific type of pyrolysis for aliphatic-based coordination frameworks, unusual for the aromatic-based MOFs. The aromatic ones usually undergo decomposition with partial conversion of the organic ligand into carbon [122,123] due to its high content and a subsequently incomplete volatilization, especially when carrying the pyrolysis out under an inert (non-oxidative) atmosphere. Moreover, a reduction of metal by carbon is possible during this process. Both problems are undesirable if the synthesis of the shaped or textured metal-oxide phase is needed, and avoiding both problems needs the pyrolysis to be carried out only in the oxidative atmosphere and at a very high temperatures (typically T > 700 °C) to provide full carbon combustion. On the contrary, the aliphatic moieties, having lower carbon content, typically decompose at significantly lower temperatures with the generation of much more volatile substances instead of carbon. Such a feature was successfully applied [124] in the synthesis of nanostructured magnesia (MgO) and ceria (CeO_2_) by a thermolysis of adipate metal–organic frameworks under an inert atmosphere. In this approach, long-chain aliphatic ligands were found to act as the self-templates, which afterwards evaporated to generate nanopores in the oxide phases. Significantly, CO_2_ adsorption measurements for the obtained nanoporous MgO exhibited exceptional adsorption capacity (9.2 wt.%) under conditions mimicking flue gas, confirming the usability of the presented approach for the generation of chemically pure and porous metal oxides under mild conditions. 

A more complex application of lower thermal stability of the aliphatic ligand was implemented in a work [125]. Mixtures of *trans*-1,4-cyclohexanedicarboxylic (H_2_chdc) and terephtalic (H_2_bdc) acids were used in the synthesis of mixed-ligand UiO-66-type frameworks. As UiO-66-chdc was shown to thermally decompose in air at ca. 275 °C, unlike UiO-66-bdc with a thermal stability up to ca. 450 °C, a controlled thermolysis of mixed-ligand samples in air at 325 °C led to a selective decomposition of chdc bridges while terephthalates still remained. The resulting controlled incorporation of the missing linker defects into the pyrolyzed UiO-66 samples was possible for up to ca. 57% substitution of bdc by chdc in the initially synthesized samples. At higher concentrations, the amorphization of the pyrolyzed structure was observed due to too low metal node connectivity appearing in the generated 3D coordination frameworks. Such an approach represented a pathway to design missing linker defects in UiO-66-type structure with controllable concentrations, at the same time surviving their fully random distribution in the bulk structure. 

### 2.6. Outlook

In addition to the properties and current applications of the aliphatic MOFs, described in the previous sections, the saturated hydrocarbon core can be apparently called more promising for the synthesis of biostable and biocompatible MOFs and MOF-derived materials compared to the aromatic backbone. This suggestion arises from such properties as low optical absorbance, providing enhanced light stability [114,115,116], and chemical inertness of the aliphatic hydrocarbons, which is apparently superior to the inertness of the aromatic ones. The persistence of coordination polymers in vital media is not equal to the hydrolytic stability and needs extensive and comprehensive investigation. For example, recent papers report an aromatic ZIF-8, a widely recognized golem of MOF stability, to be not so invincible in biomimicking systems, as well as to be considerably toxic for several cell lines [126,127]. In this regard, a classical bioisosteric concept, which is, in particular, implemented in a substitution of aromatic fragments to their saturated structural analogues for the synthesis of diverse drugs [128,129,130,131,132], is also suitable in the chemistry of MOFs and coordination polymers [99] to significantly improve their real stability for the separations, toxicant removal, optical devices, and other practical applications.

## 3. Conclusions

In summary, the alicyclic or aliphatic ligand core may provide a great row of unique properties and structural features for metal–organic frameworks. Increased hydrophobicity of such ligands may strongly affect the crystal structure of the derived coordination framework, as well as provide unique adsorption uptakes and selectivity by either higher affinity to the non-polar substrates or fine-tuning the pore shape and size. Such properties as low optical absorption and conformational breathing in polymeric porous networks are directly provided by the physical nature of the aliphatic moieties, while the latter also implies, on the one hand, a complexity of the synthesis of the related MOFs and, on the other hand, a potentially huge structural diversity. Rotational and/or torsion mobility of such ligands has been successfully used in the design of spontaneous electrization or paraelectricity of the materials, and a relatively low thermal stability of the aliphatic ligands is one of possible keys to the effectively controlled defect engineering.

The field of aliphatic-based MOFs looks not well-developed yet regarding their synthesis, while a number of coexisting phases with the close structures and compositions are often crystallized and are hard to separate. However, unique properties of such coordination frameworks have become a driving force for their future investigations. The application of modern different NMR and EPR spectroscopic techniques, in situ diffraction, EXAFS, and in situ IR/Raman spectroscopy provide very important data concerning the structure and physical nature of these MOFs in representative works and should be further developed to provide deeper insights into the formation of dynamic aliphatic MOFs, their phase transitions, and the nature and mechanisms of adsorption processes on the aliphatic surface.

## Figures and Tables

**Figure 1 polymers-15-02891-f001:**
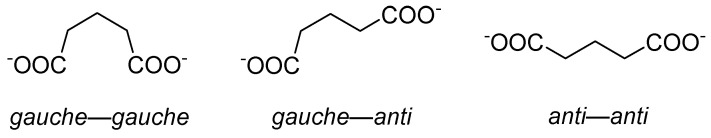
Planar projections representing three main conformations of the glutarate anion.

**Figure 2 polymers-15-02891-f002:**
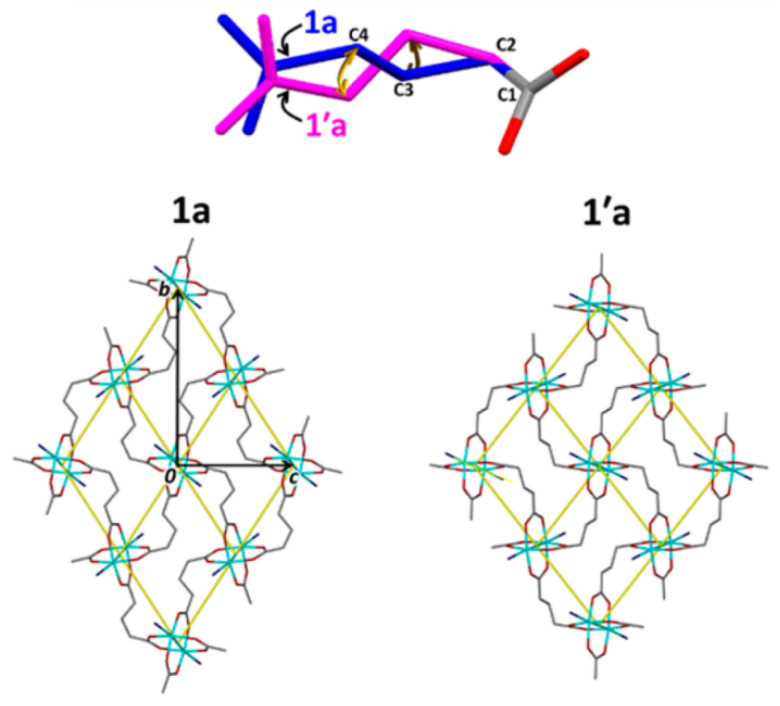
Single glutarate linker in **1a** and **1′a** (top) and a presentation on one Cu-glutarate layer in **1a** and **1′a** (bottom). Reprinted with permission from Ref. [28]. © 2017 by American Chemical Society.

**Figure 3 polymers-15-02891-f003:**
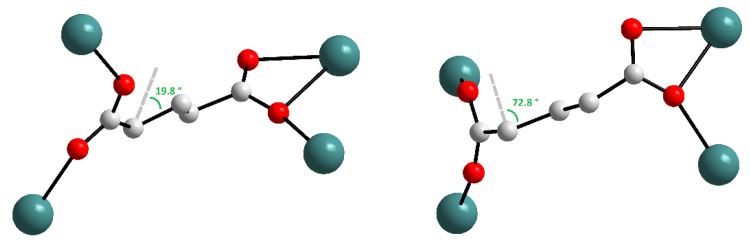
Single glutarate linker in bioMIL-2 (**left**) and bioMIL-2-hyd (**right**). Built from CCDC 775558 and 775557 entries, which were first published in Ref. [38].

**Figure 4 polymers-15-02891-f004:**
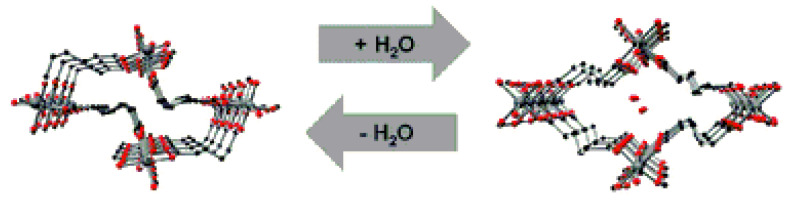
View along the channels in the activated [Al(OH)(adp)] (**left**) and the as-synthesized [Al(OH)(adp)]·H_2_O (**right**). Reprinted with permission from Ref. [51]. © 2016 by Royal Society of Chemistry.

**Figure 5 polymers-15-02891-f005:**
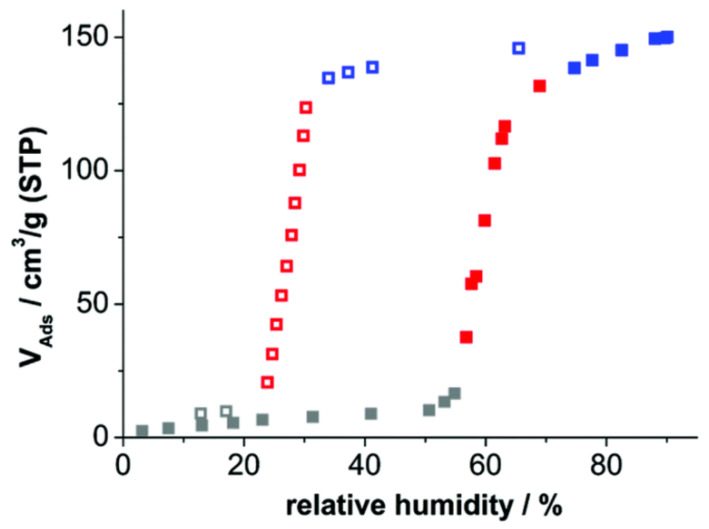
H_2_O adsorption/desorption isotherms for [Al(OH)(adp)] at 298 K. Reprinted with permission from Ref. [51]. © 2016 by Royal Society of Chemistry.

**Figure 6 polymers-15-02891-f006:**
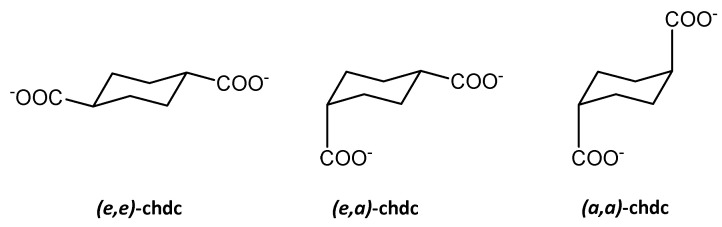
Three existing isomers of cyclohexane-1,4-dicarboxylic acid.

**Figure 7 polymers-15-02891-f007:**
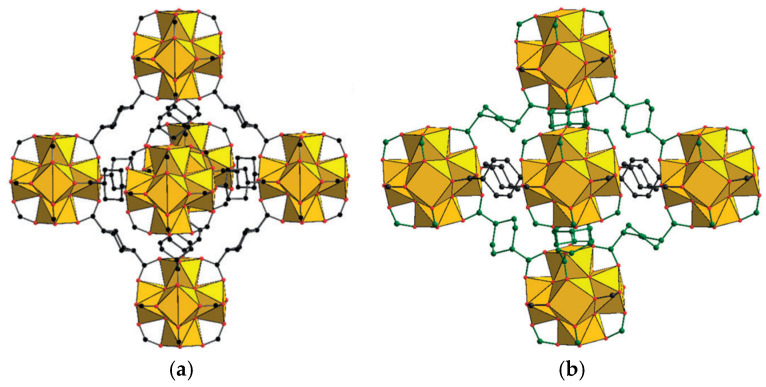
Cubic cage in the as-synthesized form of [Zr_6_O_4_(OH)_4_(chdc)_6_]·xGuest (**a**) and the modelled tetragonal cage in the activated form [Zr_6_O_4_(OH)_4_(chdc)_6_] (**b**). Reprinted with permission from Ref. [55]. © 2016 by John Wiley and Sons.

**Figure 8 polymers-15-02891-f008:**
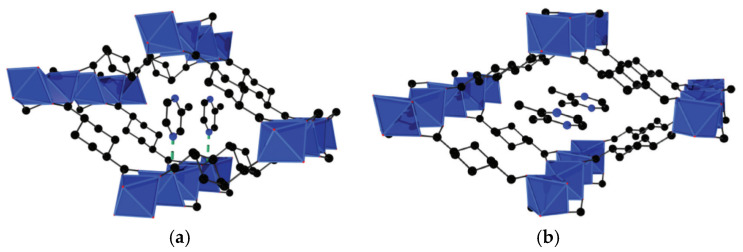
Fragments of 3D porous structure in a narrow-channel form of [Al(OH)(chdc)] filled by methylpyrazine (**a**) and an open-channel one filled by o-dimethylpyrazine (**b**). Reprinted with permission from Ref. [57]. © 2017 by Royal Society of Chemistry.

**Figure 9 polymers-15-02891-f009:**
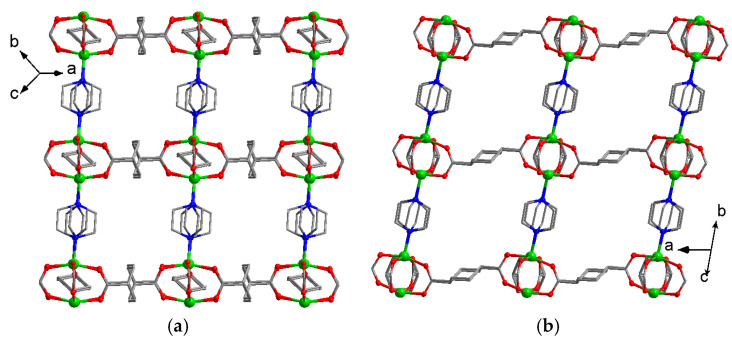
View along the channels in the as-synthesized form of [Zn_2_(chdc)_2_(dabco)]: constructed by dabco and (*a*,*a*)-chdc (**a**) and by dabco and (*e*,*e*)-chdc (**b**). Reprinted with permission from Ref. [71]. © 2020 by American Chemical Society.

**Figure 10 polymers-15-02891-f010:**
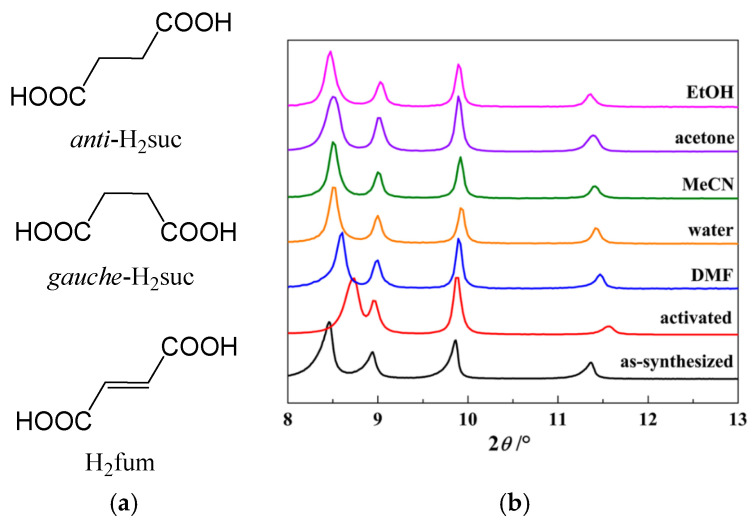
Planar structure projections of *anti*-succinate, *gauche*-succinate, and fumarate (**a**). PXRD patterns for Zr(IV)-succinate in several forms: as-synthesized, activated, and immersed in different solvents (**b**). (**b**) is reprinted with permission from Ref. [73]. © 2019 by American Chemical Society.

**Figure 11 polymers-15-02891-f011:**
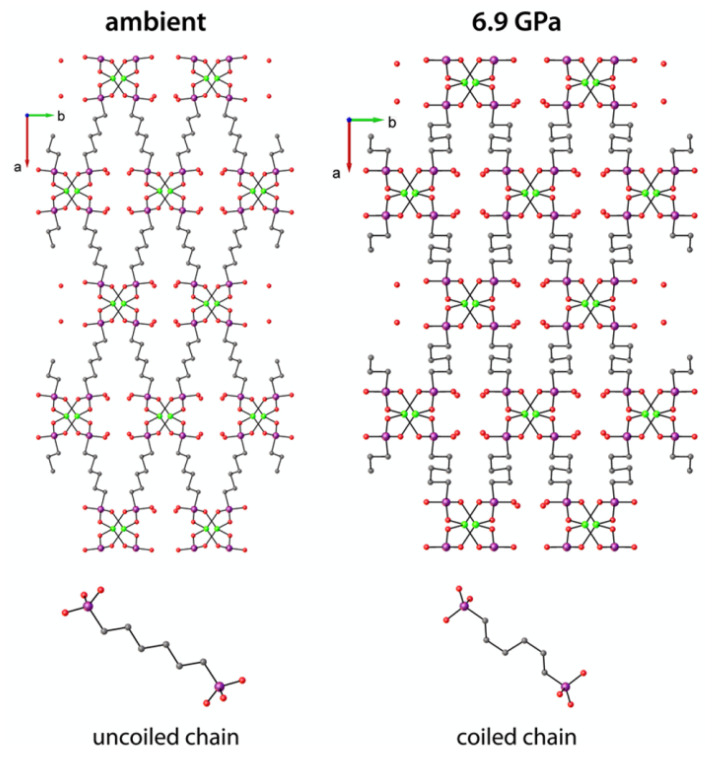
View along the *c*-axis for crystal structures for Zn-hexane-1,6-diphosphonate at ambient and high pressures (**top**) and single bridging ligand corresponding to these structures (**bottom**). Reprinted with permission from Ref. [74]. © 2014 by American Chemical Society.

**Figure 12 polymers-15-02891-f012:**
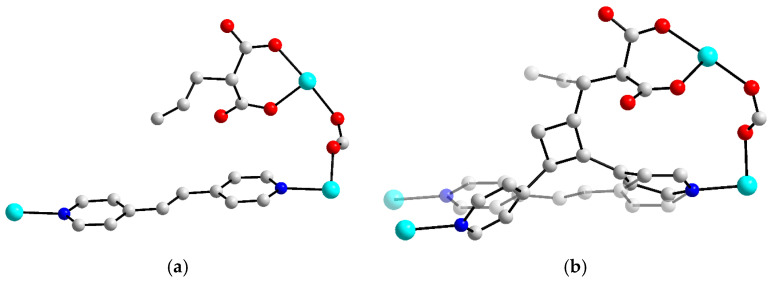
Fragments of crystal structures of Zn-bpe-allylmalonate before (**a**) and after (**b**) UV-irradiation, representing a partial [2+2] dimerization. Built from CCDC 1868491 and 1868492 entries, which were firstly published and discussed in Ref. [74].

**Figure 13 polymers-15-02891-f013:**
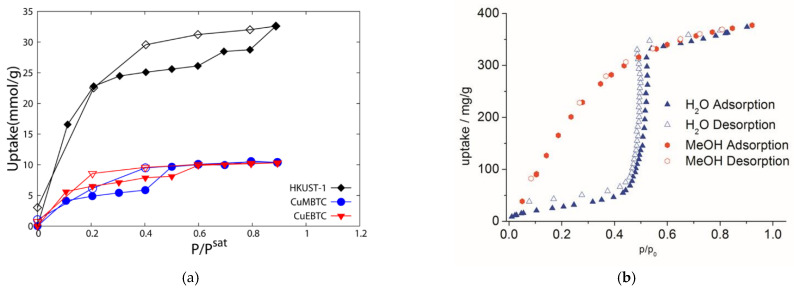
Water adsorption isotherms for HKUST-1 and its methylated and ethylated derivatives (**a**). Reprinted with permission from Ref. [90]. © 2012 by American Chemical Society. Water and methanol adsorption isotherms for aluminum mesaconate (**b**). Reprinted with permission from Ref. [91]. © 2017 by John Wiley and Sons.

**Figure 14 polymers-15-02891-f014:**
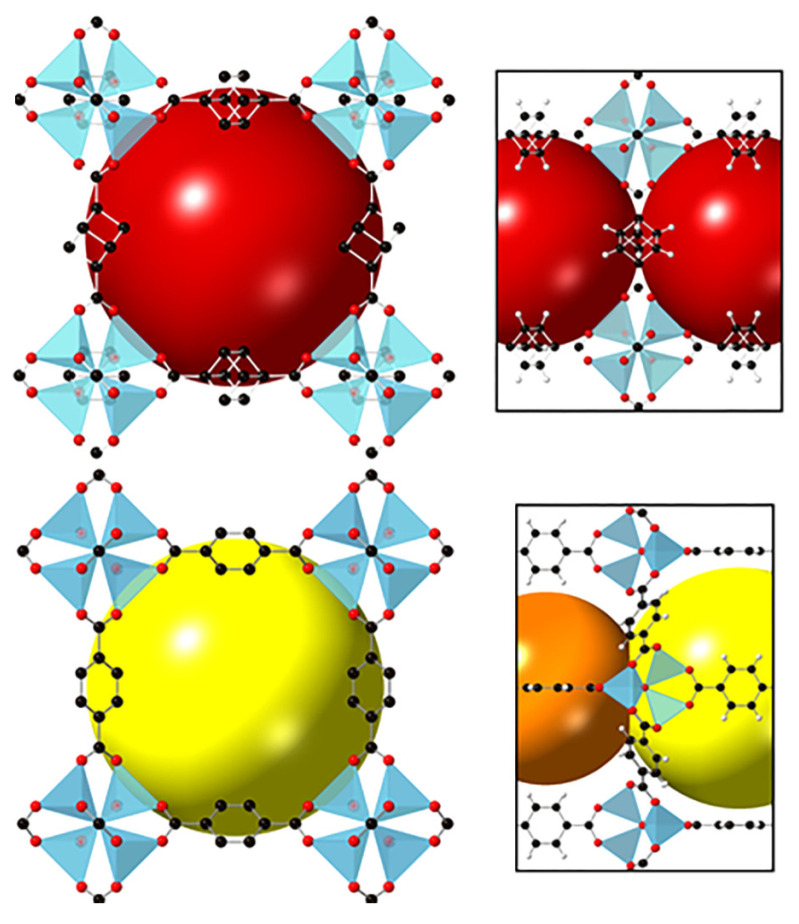
Cubic-like pores in the structures of **CUB-5** (**top**) and **MOF-5** (**bottom**). Reprinted with permission from Ref. [94]. © 2019 by American Chemical Society.

**Figure 15 polymers-15-02891-f015:**
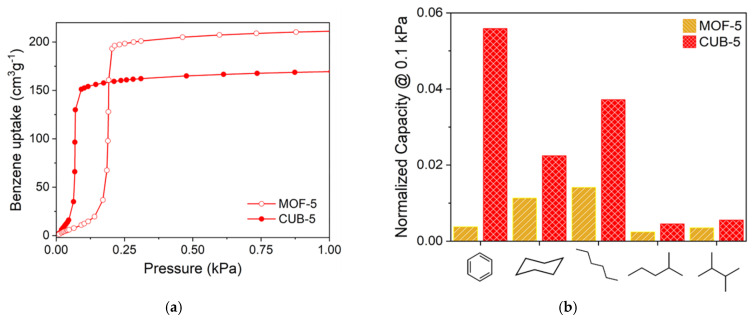
Benzene adsorption isotherms for **MOF-5** and **CUB-5** at 298 K (**a**). The relative uptakes of different hydrocarbons for **MOF-5** and **CUB-5** (**b**) at 0.1 kPa. Reprinted with permission from Ref. [94]. © 2019 by American Chemical Society.

**Figure 16 polymers-15-02891-f016:**
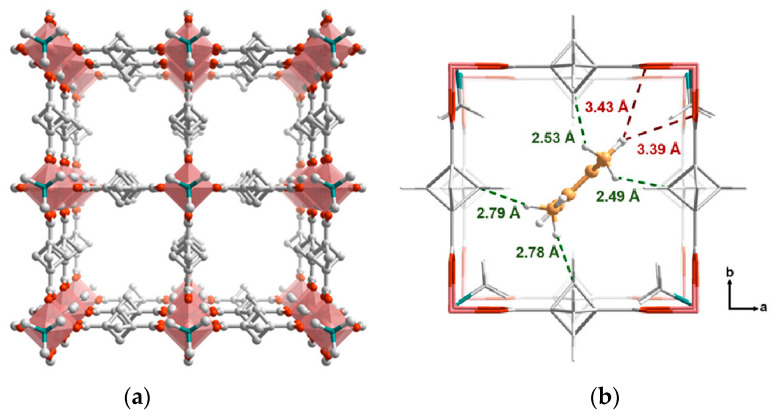
Three-dimensional structure of **ZUL-C3**, view along the channels (**a**). DFT-optimized position of o-xylene molecules in the pore with the shortest host–guest contacts illustrated (**b**). Reprinted with permission from Ref [96]. © 2022 by American Chemical Society.

**Figure 17 polymers-15-02891-f017:**
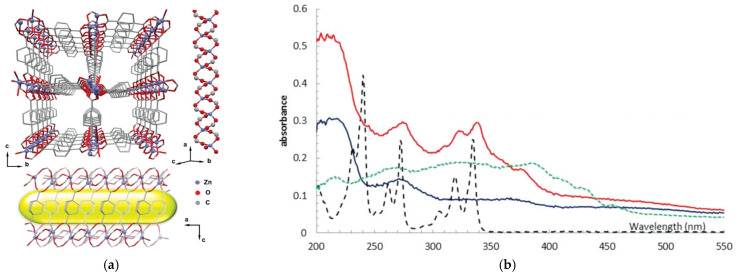
Channels in the structure of TMOF, view along the *a-* and *b*-axis (**a**). UV/vis diffuse reflectance spectra of activated TMOF (blue), activated pyrene@TMOF (red), solid pyrene (green), and 10^−4^ M ethanol solution of pyrene (black dashed) (**b**). Reprinted with permission from Ref. [102]. © 2017 by Royal Society of Chemistry.

**Figure 18 polymers-15-02891-f018:**
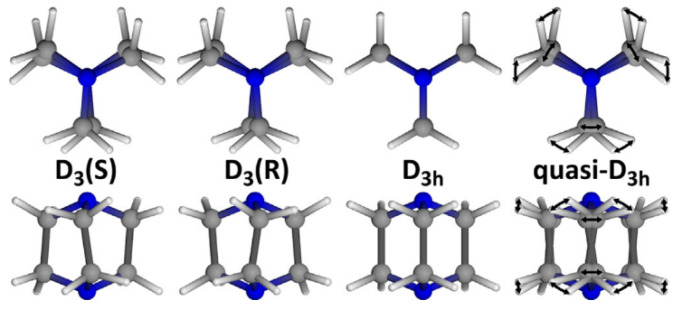
Possible conformations of the dabco molecule. Reprinted with permission from Ref. [117]. © 2018 by Elsevier.

**Figure 19 polymers-15-02891-f019:**
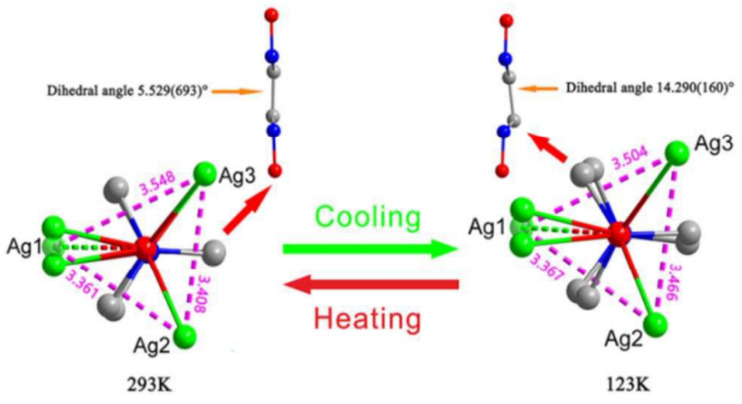
Representation of odabco distortion in [Ag_3_(odabco)(NO_3_)_3_]·H_2_O at phase transitions. Reprinted with permission from Ref. [118]. © 2017 by Royal Society of Chemistry.

**Table 1 polymers-15-02891-t001:** Simplest linear aliphatic dicarboxylic acids and a number of their derived crystal structures of coordination polymers.

Acid Name	Structural Formula	Coordination Polymers with Unsubstituted Ligand *	Coordination Polymers with Any Substituted H Atom in the Ligand *
Malonic	HOOC–CH_2_–COOH	333	685
Succinic	HOOC–(CH_2_)_2_–COOH	667	2638
Glutaric	HOOC–(CH_2_)_3_–COOH	409	1873
Adipic	HOOC–(CH_2_)_4_–COOH	411	1520
Pimelic	HOOC–(CH_2_)_5_–COOH	99	619
Suberic	HOOC–(CH_2_)_6_–COOH	72	354
Azelaic	HOOC–(CH_2_)_7_–COOH	52	277
Sebacic	HOOC–(CH_2_)_8_–COOH	43	57
Terephthalic **	HOOC–(C_6_H_4_)–COOH	3201	9881

* According to Cambridge Structural Database [21] 5.44 (April 2023) structure search. ** The data for simplest aromatic dicarboxylic acid is provided for comparison.

**Table 2 polymers-15-02891-t002:** The most common alicyclic polycarboxylic acids and a number of their derived crystal structures of coordination polymers.

Acid Name *	Skeletal Formula	Coordination Polymers with Unsubstituted Ligand **
Cyclobutane-1,1-dicarboxylic	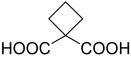	66
Cyclohexane-1,2-dicarboxylic	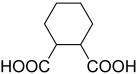	104
Cyclohexane-1,4-dicarboxylic	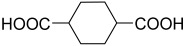	354
Cyclohexane-1,3,5-tricarboxylic	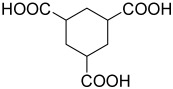	56
Cyclohexane-1,2,4,5-tetracarboxylic	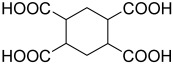	54
Cyclohexane-1,2,3,4,5,6-hexacarboxylic	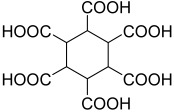	59
Adamantane-1,3-dicarboxylic	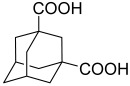	114
Camphoric	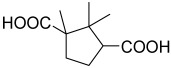	257

* Any diastereomers, enantiomers and conformers are included. ** According to Cambridge Structural Database [21] 5.44 (April 2023) structure search.

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
