# Peer review of "Properties of Aliphatic Ligand-Based Metal–Organic Frameworks"

_polymers, 2023, doi:10.3390/polym15132891_

Round 1

Reviewer 1 Report

Ligands with a purely aliphatic backbone receive a rising attention in the chemistry of coordination polymers and metal-organic frameworks. A number of unique features inherent to the aliphatic bridges, such as increased conformational freedom, non-polarizable core and low lightabsorption provide rare and valuable properties for their-derived MOFs. Applications of such compounds in stimuli-responsive materials, gas and vapor adsorbents with high and unusual selectivities, light-emitting and optical materials has extensively emerged in recent years. These properties, as well as other specific features of aliphatic-based metal-organic frameworks are summarized and analyzed in this short critical review authored by Demakov.

This review is well organized, so I recommend it to be published in Polymers as it is withour futher review.

Author Response

Thank you for your kindly high evaluation of the review. 

Reviewer 2 Report

The review paper entitled "Properties of aliphatic ligand-based metal-organic frameworks" by P.A. Demakov highlights the state-of-the-art of MOFs built up by using flexible aliphatic organic ligands. The review sum up a number of 123 references, a part belonging to the author. Overall, the composition of the review is well structured, being emphasized the main properties of the flexible MOFs: breathing behavior depending on the conformation of the ligands used in the construction of MOFs, adsorption properties as a particular behavior of flexible MOFs, optical and thermal characteristics as a result of the coordination of different metal centers.

In general, the chosen literature is relevant for this category of MOFs, I have the following suggestions:

In keywords: the word "coordination polymers" is used only twice (lines 8 and 37) so that it can not be considered as a keyword in this case.

In Introduction section: it is necessary to add  a table including the main class of the aliphatic acids used as ligands for MOF synthesis, as well as their mixtures with ancillary ligands.

The Section 2 related to the properties of the aliphatic-based MOFs is well presented. The missing part is that related to the hydrolytic stability of the aliphatic-based MOFs, and the advantages reported to the aromatic-based MOFs.

Before Conclusions Section some Perspectives and current application, as well as the challenges of the development of aliphatic MOFs must be added.

Based on these observations, my recommendation if Minor revision before the publication.  

Author Response

Thank you for the careful evaluation of the manuscript. Please see the attachemt. 

Reviewer 3 Report

The author present a review of aliphatic metal-organic frameworks and their properties, emphasizing when possible, their effect regarding linker flexibility. The review presented is sound and I believe will have a good impact on a area of MOFs not well explored. 

Overall, the writing is good and sound. A more attentive reading is needed just to correct some writing oversights.

Author Response

Thank you for your kindly high evaluation of the review. English grammar and style have been carefully revised.